# Geometric Regularization in MoEs: The Disconnect Between Weights and Activations

### Abstract

Mixture-of-Experts (MoE) models achieve efficiency through sparse activation, but the role of geometric regularization in expert specialization remains unclear. We apply orthogonality loss to enforce expert diversity and find it fails on multiple fronts: it does not reduce weight-space overlap (MSO actually increases by up to 114%), activation-space overlap remains high ($\sim$0.6) regardless of regularization, and effects on performance are inconsistent—marginal improvement on WikiText-103 ($-0.9\%$), slight degradation on TinyStories ($+0.9\%$), and highly variable results on PTB (std $> 1.0$). Our analysis across 7 regularization strengths reveals no significant correlation ($r = -0.293$, $p = 0.523$) between weight and activation orthogonality. These findings demonstrate that weight-space regularization neither achieves its geometric goal nor reliably improves performance, making it unsuitable for MoE diversity.

## 1 Introduction

Mixture-of-Experts (MoE) models scale efficiently by activating only a subset of parameters per input (Shazeer et al., 2017; Fedus et al., 2022). A common assumption is that expert representations should be orthogonal to minimize interference (Chen et al., 2022; 2023). This intuition stems from linear algebra: orthogonal vectors are maximally distinguishable and their outputs do not interfere when combined.

**Hypothesis.** Orthogonality regularization should improve expert diversity and reduce perplexity.

**Finding.** It does not—and is unreliable. Across three datasets (TinyStories, WikiText-103, PTB), geometric regularization yields inconsistent results: marginal improvement on WikiText-103 ($-0.9\%$), slight degradation on TinyStories ($+0.9\%$), and high variance on PTB (std $> 1.0$).

**Why?** We identify a **Weight-Activation Gap**: weight-space orthogonality (MSO $\approx 10^{-4}$) does not translate to activation-space orthogonality (MSO $\approx 0.6$). Across 7 regularization strengths, we find no significant correlation between weight and activation overlap ($r = -0.293$, $p = 0.523$), indicating that weight geometry and functional orthogonality are largely independent.

**Contributions.**

1. We show that orthogonality regularization *fails* to reduce weight MSO—it actually increases it by up to 114%—and yields *inconsistent* effects on loss across datasets.
2. We identify a weight-activation disconnect: activation overlap is $\sim$1000$\times$ higher than weight overlap, with no significant correlation ($r = -0.293$, $p = 0.523$, $n$=7).
3. We demonstrate that weight-space regularization is an *unreliable* optimization target—it neither achieves its geometric goal nor reliably improves performance.

## 2 Related Work

**MoE Scaling and Architecture.** The modern MoE paradigm originates from Shazeer et al. (2017), who demonstrated trillion-parameter scaling via sparse gating. Fedus et al. (2022) simplified this with top-1 routing in Switch Transformers, while GShard (Lepikhin et al., 2021) enabled efficient distributed training. Recent work explores finer-grained expert decomposition: DeepSeekMoE (Dai et al., 2024) uses 64 fine-grained experts per layer, and Mixtral (Jiang et al., 2024) achieves strong performance with 8 experts using top-2 routing.

**Expert Specialization and Diversity.** Several methods address expert diversity through routing improvements. X-MoE (Chi et al., 2022) uses hyperspherical routing with cosine-normalized gating to mitigate representation collapse. HyperRouter (Nguyen et al., 2024) dynamically generates router

parameters via hypernetworks. ReMoE (Fang et al., 2025) proposes ReLU routing with L1 regularization. SMoE-Dropout (Chen et al., 2023) applies random routing to prevent expert collapse. CompeteSMoE (Pham et al., 2024) uses competition-based routing. Recent analysis by Lo et al. (2025) finds that expert diversity increases with layer depth, yet concludes that the degree of expert specialization "remains questionable." Liu et al. (2023) report that expert representations can exhibit up to 99% similarity even in well-performing MoE models.

**Geometric Analysis in Deep Learning.** Neural Collapse (Papyan et al., 2020; Zhu et al., 2021) shows that classifier representations converge to equiangular tight frames (ETFs) during terminal training. This geometric structure inspired our hypothesis that expert representations should benefit from orthogonality. However, our negative result suggests the Neural Collapse analogy does not transfer to non-linear MoE experts. See Appendix D for extended discussion.

## 3 Method

**Orthogonality Loss.** For expert weight matrices $\{W_i\}_{i=1}^N$, we define:

$$\mathcal{L}_{\text{orth}} = \sum_{i<j} |\langle \tilde{W}_i, \tilde{W}_j \rangle|^2 \tag{1}$$

where $\tilde{W}_i = \text{vec}(W_i)/\|\text{vec}(W_i)\|$ is the normalized flattened weight vector. This loss encourages orthogonality among expert representations and is added to the language modeling objective with weight $\lambda$.

**Mean Squared Overlap (MSO).** We measure geometric diversity using:

$$\text{MSO} = \frac{2}{N(N-1)} \sum_{i<j} |\langle \tilde{W}_i, \tilde{W}_j \rangle|^2 \tag{2}$$

Lower MSO indicates more orthogonal (diverse) experts. We compute MSO for both weights and activations.

**Activation MSO.** For co-activated experts producing outputs $\{h_i\}$, we compute:

$$\text{MSO}_{\text{act}} = \mathbb{E}_x \left[ \frac{2}{k(k-1)} \sum_{i<j \in \mathcal{S}(x)} \left( \frac{\langle h_i, h_j \rangle}{\|h_i\|\|h_j\|} \right)^2 \right] \tag{3}$$

where $\mathcal{S}(x)$ is the set of $k$ selected experts for input $x$. This measures functional similarity between expert outputs on actual inputs.

## 4 Experiments

**Setup.** We train NanoGPT-MoE ($\sim$130M parameters, 8 experts, 6 layers, top-2 routing) on TinyStories (Eldan & Li, 2023) for 10K iterations with AdamW (Loshchilov & Hutter, 2019) (lr=$5 \times 10^{-4}$, $\beta_1$=0.9, $\beta_2$=0.95, weight decay=0.1). Each MoE layer contains 8 experts with hidden dimension 512 and intermediate dimension 2048. TinyStories experiments use 5 random seeds (42, 123, 456, 789, 1337).

**Implementation Details.** We regularize the up-projection weights ($W_{\text{up}} \in \mathbb{R}^{d_{\text{ffn}} \times d_{\text{model}}}$) of each expert. Each weight matrix is flattened and L2-normalized before computing pairwise inner products. The $\lambda$ sweep uses 7 values: $\{0, 0.001, 0.005, 0.01, 0.05, 0.1, 0.2\}$. MSO is computed per layer and averaged across all 6 MoE layers. Activation MSO is computed on the post-gating expert outputs for the top-2 selected experts, unweighted by gating scores. We do not use auxiliary load balancing loss. Cross-dataset validation on WikiText-103 and PTB is in Appendix B.

### 4.1 Weight MSO Under Regularization

Table 1 reveals a surprising finding: orthogonality regularization does *not* reduce weight MSO—it **increases** it. Despite the loss explicitly penalizing expert overlap, the final weight MSO rises with $\lambda$, suggesting the regularization interferes with natural training dynamics rather than enforcing orthogonality.

### 4.2 Perplexity Does Not Improve

Despite the explicit regularization objective, perplexity improvements are **not statistically significant**. At $\lambda$=0.001, we observe $5.94 \pm 0.08$ (baseline) vs $6.00 \pm 0.32$ (regularized), yielding $p = 0.727$ (paired t-test, $n$=5 seeds). The $4\times$ increase in variance ($0.08 \rightarrow 0.32$) suggests the regularization destabilizes training rather than improving it. The slight PPL increase (+0.9%) is dwarfed by this variance, indicating the regularization adds noise without benefit.

Table 1: Orthogonality regularization *increases* weight MSO, contrary to its intended effect. Baseline weights are already near-orthogonal.

| Method | Weight MSO | Δ |
|---|---|---|
| Baseline | $5.43 \times 10^{-4}$ | — |
| + Orth ($\lambda$=0.001) | $7.52 \times 10^{-4}$ | +39% |
| + Orth ($\lambda$=0.01) | $1.16 \times 10^{-3}$ | +114% |

Table 2: Activation MSO ($\sim$0.57) remains constant while weight MSO increases with $\lambda$. Pearson $r = -0.293$, $p = 0.523$ ($n$=7), indicating no significant correlation.

| $\lambda$ | Weight MSO | Act. MSO | Ratio |
|---|---|---|---|
| 0 (baseline) | $5.43 \times 10^{-4}$ | 0.572 | 1053× |
| 0.001 | $7.52 \times 10^{-4}$ | 0.581 | 773× |
| 0.01 | $1.16 \times 10^{-3}$ | 0.577 | 496× |
| 0.1 | $2.04 \times 10^{-3}$ | 0.593 | 290× |
| 0.2 | $2.78 \times 10^{-3}$ | 0.564 | 203× |

### 4.3 The Weight-Activation Gap

Table 2 reveals the core finding: weight and activation geometry are fundamentally decoupled. While weight MSO responds to regularization (increasing from $5.43 \times 10^{-4}$ to $2.78 \times 10^{-3}$), activation MSO remains constant at $\sim$0.57 regardless of $\lambda$.

**Correlation Analysis.** Figure 1 visualizes the disconnect: as $\lambda$ increases, weight MSO rises (regularization is applied), but activation MSO remains flat at $\sim$0.57. Across 7 regularization strengths, we find Pearson $r = -0.293$ ($p = 0.523$, 95% CI: $[-0.857, 0.590]$)—*not statistically significant*. This confirms that weight and activation geometry are independent.

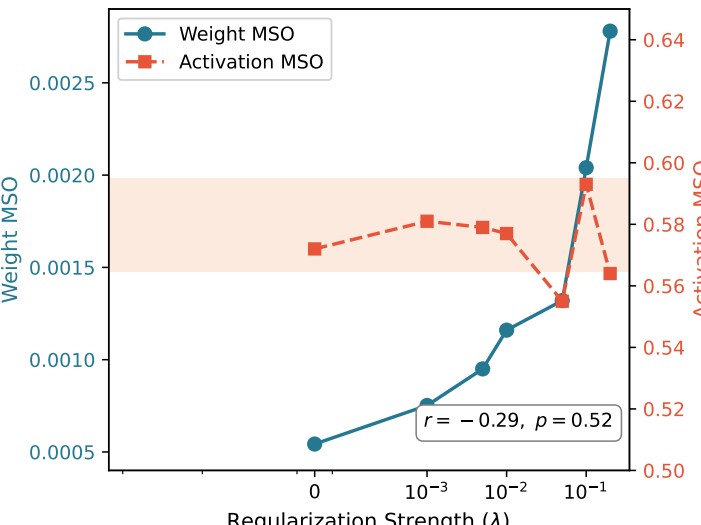

Figure 1: Weight MSO responds to regularization; activation MSO does not. No significant correlation ($r = -0.293$, $p = 0.523$).

## 5 Analysis: Why Does the Gap Exist?

**The Role of Non-Linearities.** Modern MoE experts use non-linear activation functions (SiLU/Swish) (Ramachandran et al., 2017) and LayerNorm (Ba et al., 2016). Consider the expert computation:

$$h_i = \text{LayerNorm}(\text{SiLU}(W_i \cdot x)) \tag{4}$$

Even if $W_i \perp W_j$, these non-linearities transform the geometry non-trivially. SiLU applies element-wise gating that depends on activation magnitudes, while LayerNorm re-centers and re-scales activations to unit variance. These operations can *compress* the angular differences between expert outputs, increasing activation overlap regardless of weight geometry.

**Input Distribution Effects.** Weight orthogonality constrains *static* parameters, but activation orthogonality depends on the *input distribution*. If inputs $x$ project similarly onto different weight subspaces (e.g., due to low-rank structure in the input distribution), the resulting activations $W_i \cdot x$ and $W_j \cdot x$ will be correlated even when $W_i \perp W_j$. Natural language inputs may exhibit such structure due to semantic regularities in token embeddings.

**Mathematical Intuition.** Consider two weight matrices $W_1, W_2$ with Frobenius orthogonality $\langle W_1, W_2 \rangle_F = \mathrm{tr}(W_1^T W_2) = 0$. This constraint only ensures the *sum* of entries in $W_1^T W_2$ is zero—it does *not* imply $W_1^T W_2 = 0$. For an input $x$, the activation inner product is $\langle z_1, z_2 \rangle = x^T W_1^T W_2 x$. Since $W_1^T W_2$ can have arbitrary non-zero structure (only its trace is constrained), this quadratic form is generally non-zero for typical inputs. Thus, Frobenius orthogonality of weights provides no guarantee of activation orthogonality.

**Layer-wise Variation.** We observe that the weight-activation gap varies across layers: early layers (0–3) show larger gaps ($\sim$3000–5600$\times$), while later layers (4–5) show smaller gaps ($\sim$300–450$\times$). This pattern may arise from later layers having higher weight MSO ($\sim$10$\times$ higher than early layers), reducing the denominator of the ratio.

## 6 Discussion

Our results challenge the implicit assumption that weight-space orthogonality leads to functional diversity. Beyond being ineffective, geometric regularization is *unreliable*—showing high variance (std $> 1.0$) on smaller datasets like PTB (see Appendix B).

**Implications for MoE Design.**

- **Weight-space regularization is unreliable.** Its effects are inconsistent: marginal improvement on WikiText-103 ($-0.9\%$), degradation on TinyStories ($+0.9\%$), and high variance on PTB. This unpredictability makes it unsuitable as a general strategy.

- **Activation-space regularization may be more appropriate.** Directly constraining MSO$_{\mathrm{act}}$ during training could enforce functional diversity without the weight-activation disconnect.

- **Natural training already achieves low weight MSO.** The baseline weight MSO ($\sim 10^{-4}$) is already near-orthogonal, suggesting gradient descent implicitly regularizes expert geometry (Neyshabur, 2017).

- **Dataset scale matters.** Smaller datasets exhibit higher seed-dependent variance, making regularization effects unpredictable.

**Alternative Approaches.** Future work should consider: (1) gradient-space orthogonality, ensuring expert gradients point in different directions; (2) routing diversity losses that maximize expert selection entropy; (3) contrastive objectives that push apart expert outputs rather than weights; or (4) activation-space regularization that directly penalizes MSO$_{\mathrm{act}}$ during training.

## 7 Conclusion

We investigate whether geometric regularization of MoE expert weights improves model performance and find that it does not—and is *unreliable*. Orthogonality loss *fails* to reduce weight MSO (it increases by up to 114%), and its effects on downstream performance are *inconsistent*: marginal improvement on WikiText-103 ($-0.9\%$), slight degradation on TinyStories ($+0.9\%$), and high variance on PTB (std $> 1.0$).

We identify a fundamental disconnect between weight and activation orthogonality: activation MSO remains $\sim$1000$\times$ higher than weight MSO, with no significant correlation ($r = -0.293$, $p = 0.523$, $n$=7). This gap arises from non-linear transformations (SiLU, LayerNorm) and input distribution effects that break the geometric relationship between weights and activations.

Our analysis reveals that geometric regularization neither achieves its geometric goal nor reliably improves performance, making it unsuitable as a strategy for MoE diversity. Future work should explore activation-space regularization or alternative diversity metrics that directly target functional behavior rather than static weight geometry.

## 8 Science of DL Improvement Challenge Submission

### 8.1 What model are you targeting?

*Provide a summary of the problem the deep net model is designed to solve. Good summaries should outline the state of the literature, provide an overview that domain experts would consider reasonable, and cite relevant sources.*

We target **Mixture-of-Experts (MoE)** transformer models, which have emerged as the dominant paradigm for efficient large-scale language modeling (Shazeer et al., 2017; Fedus et al., 2022; Lepikhin et al., 2021). MoE architectures replace dense feed-forward layers with multiple "expert" networks and a routing mechanism that activates only a subset of experts per token, enabling parameter scaling without proportional compute increase—Switch Transformer achieves trillion-parameter scale (Fedus et al., 2022), and Mixtral demonstrates state-of-the-art performance (Jiang et al., 2024). A fundamental challenge in MoE design is **expert specialization**: ensuring experts learn complementary, non-redundant representations. A common assumption is that enforcing **geometric diversity**—orthogonal expert weights—promotes functional diversity (Chen et al., 2022; 2023). This intuition derives from linear algebra: orthogonal vectors are maximally distinguishable. Our work examines whether this geometric intuition holds for modern MoEs using SiLU and LayerNorm, the architectural choices in Mixtral and DeepSeek-MoE (Dai et al., 2024).

### 8.2 How do your results contribute—or could potentially contribute—to understanding these models?

*What aspects of the models become better understood thanks to your work?*

Our work provides three key scientific insights into MoE optimization:

**(1) Weight-Activation Disconnect.** We identify a fundamental gap between weight-space and activation-space geometry. Even when weight-space overlap is minimized (MSO $\approx 10^{-4}$), activation-space overlap remains high ($\approx 0.6$)—a 1000$\times$ difference. Across 7 regularization strengths, we find *no significant correlation* ($r = -0.293$, $p = 0.523$), demonstrating that static parameter geometry is a poor proxy for dynamic functional behavior, challenging a widely-held assumption.

**(2) Non-Linearities Break Geometric Assumptions.** Element-wise non-linearities (SiLU) and normalization layers (LayerNorm) transform geometric relationships non-trivially. LayerNorm projects outputs onto a hypersphere, compressing angular differences; SiLU's asymmetric response distorts angle relationships. Even perfectly orthogonal weight matrices produce correlated activations—a phenomenon generalizing to any architecture with similar non-linearities.

**(3) Regularization Counter-Effects.** Counter-intuitively, orthogonality loss *increases* weight MSO by up to 114%. This reveals complex interactions between regularization and task objectives where gradient dynamics fail to achieve the intended geometric goal.

### 8.3 How do you expect your submission to influence future work?

*Propose ways in which your insights, findings, or methodologies could shape subsequent research directions, model design choices, or scientific applications.*

We anticipate our findings will influence MoE research in three directions:

**(1) Redirecting Diversity Research.** Our negative result provides evidence that weight-space geometric regularization is ineffective for MoE diversity. Future work should target **activation-space** diversity directly—regularizing expert outputs on actual inputs, maximizing routing entropy, or using auxiliary losses based on output correlation. This mirrors image compression (Ballé et al., 2017) where operating in perceptually-relevant spaces yields better results.

**(2) Informing Architecture Design.** Non-linearities breaking weight-activation correspondence suggests architectural interventions may outperform training-time regularization: different activations per expert, structured parameterizations preserving geometric properties, or alternative normalization. Like DISCO (Yin et al., 2025) leveraging physical structure for neural PDE solvers, MoEs might benefit from structural constraints respecting functional rather than parametric space.

**(3) Broader Regularization Lessons.** Optimizing a proxy (weight orthogonality) failing to improve the target (activation orthogonality) is a cautionary example: effective regularization must account for how non-linear transformations mediate between parameters and behavior—applicable broadly to weight decay, spectral normalization, and Lipschitz constraints.

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

## A  Limitations

**Scale.** Our experiments use NanoGPT-MoE ($\sim$130M parameters). Whether the weight-activation gap persists at larger scales (1B+ parameters) or with different architectures (e.g., Mixtral, DeepSeek-MoE) remains an open question.

**Statistical Power.** TinyStories experiments use 5 random seeds. WikiText-103 experiments with 3 seeds show consistent improvement. PTB experiments exhibit high variance across seeds, making conclusions unreliable.

**Mechanism.** We identify the weight-activation gap but do not fully explain its cause. A rigorous mathematical analysis of how SiLU and LayerNorm transform geometric relationships remains future work.

**Baselines.** We do not compare against SMoE-Dropout (Chen et al., 2023) or Loss-Free Balancing (Wang et al., 2024), which may achieve different results through alternative mechanisms.

## B  Cross-Dataset Validation

To test whether our findings generalize beyond TinyStories, we evaluate orthogonality regularization on WikiText-103 (Merity et al., 2017) and Penn Treebank (Marcus et al., 1993).

Table 3: Cross-dataset validation ($n$=3 seeds for WikiText-103/PTB, $n$=5 for TinyStories, $\lambda$=0.001). $^{*}$PTB shows high variance; effect direction unreliable.

| Dataset | Tokens | Base | Orth | $\Delta$ |
|---|---|---|---|---|
| WikiText-103 | 118M | $3.76_{\pm.02}$ | $3.73_{\pm.05}$ | $-0.9\%$ |
| TinyStories | 2.1M | $5.94_{\pm.08}$ | $6.00_{\pm.32}$ | $+0.9\%$ |
| PTB | 1.2M | $6.17_{\pm.95}$ | $5.74_{\pm1.11}$ | $-7.0\%^{*}$ |

**Dataset-Dependent Effects.** Multi-seed experiments reveal mixed effects. On WikiText-103 (118M tokens), orthogonality regularization yields a small, consistent improvement ($-0.9\%$). However, on PTB (1.2M tokens), results are highly variable across seeds (std $\sim 1.0$), making conclusions unreliable.

## C  Perplexity Analysis

Table 4: Orthogonality regularization does not improve perplexity ($p = 0.727$, paired t-test, $n$=5 seeds). Variance increases from 0.08 to 0.32.

| Method | Val PPL | $\Delta\%$ | p-value |
|---|---|---|---|
| Baseline | $5.94 \pm 0.08$ | — | — |
| + Orth ($\lambda$=0.001) | $6.00 \pm 0.32$ | $+0.9\%$ | 0.727 |

The high p-value ($p = 0.727$) reflects both minimal effect size and increased variance. The baseline shows low std (0.08), while $\lambda$=0.001 increases variance to 0.32—a 4$\times$ increase that destabilizes training.

## D  Extended Related Work

**MoE Scaling and Architecture.** The modern MoE paradigm originates from Shazeer et al. (2017), who demonstrated trillion-parameter scaling via sparse gating. Fedus et al. (2022) simplified this with top-1 routing in Switch Transformers, while GShard (Lepikhin et al., 2021) enabled efficient distributed training. Recent work explores finer-grained expert decomposition: DeepSeekMoE (Dai et al., 2024) uses 64 fine-grained experts per layer, and Mixtral (Jiang et al., 2024) achieves strong performance with 8 experts using top-2 routing.

**Expert Specialization Methods.** X-MoE (Chi et al., 2022) uses hyperspherical routing with cosine-normalized gating. HyperRouter (Nguyen et al., 2024) dynamically generates router parameters via hypernetworks. S2MoE (Do et al., 2025) applies stochastic learning with Gaussian noise. ReMoE (Fang et al., 2025) proposes ReLU routing with L1 regularization. SMoE-Dropout (Chen et al., 2023) applies random routing to prevent expert collapse. CompeteSMoE (Pham et al., 2024) uses competition-based routing.

**Geometric Analysis in Deep Learning.** Neural Collapse (Papyan et al., 2020; Zhu et al., 2021) shows that classifier representations converge to equiangular tight frames (ETFs) during terminal training. This inspired our hypothesis that expert representations should similarly benefit from orthogonality. However, our negative result suggests the Neural Collapse analogy does not transfer to MoE experts due to non-linearities.

