# OpenReview forum: "Geometric Regularization in MoEs: The Disconnect Between Weights and Activations"
_ICLR.cc/2026/Workshop/Sci4DL — Submitted to Sci4DL 2026_

### Official Review · Reviewer_uErQ · 2026-02-25

**Fit:** 3
**Significance:** 2
**Confidence:** 2

**Summary:**

This manuscript investigates the effectiveness of weight-space orthogonal regularization in Mixture-of-Experts (MoE) models. The authors challenge the common assumption that enforcing diversity in expert weights is sufficient to achieve functional diversity. Through systematic experiments across multiple datasets, they identify a significant “weight–activation gap” that warrants further exploration.

**Strengths:**

1. One aspect I particularly appreciate is that the paper provides a highly valuable negative result, challenging established practices in the field. This aligns well with the expectations of sci4dl, as it raises new and meaningful questions about prevailing assumptions.

2. The authors conduct a comprehensive parameter sweep over the regularization strength, resulting in sufficiently thorough and well-supported experimental comparisons.

3. The paper offers constructive guidance by suggesting that future research should shift its focus from weight-space regularization to activation-space regularization.

**Suggestions:**

1. The authors suggest that activation-space regularization could serve as a better alternative. However, I do not understand why they did not implement and benchmark such a method themselves. In my view, this would represent a highly valuable scientific process: presenting a negative result, explaining the underlying reasons, proposing a new method, and empirically validating its effectiveness. The absence of this final step significantly weakens the overall contribution of the paper. I strongly encourage the authors to include such experiments.

2. The experiments are primarily limited to small-scale models. Not all conclusions drawn from small models necessarily generalize to larger ones. Nevertheless, I understand this limitation, as scaling experiments to large models requires substantial computational resources.

---

### Official Review · Reviewer_JEvQ · 2026-02-25

**Fit:** 2
**Significance:** 2
**Confidence:** 2

**Summary:**

The paper investigates orthogonalisation of experts and finds that an orthogonality loss leads to counter-intuitive behaviour.

**Strengths:**

mostly well-written, easy to follow, flows well

**Suggestions:**

The conclusions in the paper are perhaps formulated a bit too strong. To claim "fundamentally decoupled" requires more experiments. A plot over training epochs would be nice to see. MSO should be introduced earlier, it is used in the abstract & intro but only introduced in section 3. PPL is never introduced. I think less brackets in the abstract would look better.

---

### Official Review · Reviewer_YzRu · 2026-02-25

**Fit:** 1
**Significance:** 1
**Confidence:** 2

**Summary:**

The paper reports negative results on the effect of reducing weight-space overlap to improve MoE diversity.

**Strengths:**

The entire study looks AI generated. The motivation does not really make sense, which results in experiments not providing any meaningful insights.

**Suggestions:**

The motivation should be clearer to improve the chances of the downstream experiments to be interesting.

---

### Meta-Review · Area_Chair_WK26 · 2026-02-28

**Recommendation:** Reject

**Metareview:**

This is primarily a DL engineering paper, not a DL science paper. While there are technically empirical results reported, they are very coarse (e.g. scalar correlations). The "weight orthogonality doesn't imply activation orthogonality" message is already apparent from what we know theoretically about forward prop in deep nets (e.g. the NNGP perspective), and this null result isn't really that interesting. I don't think this is suitable for this workshop.

---

### Decision · Program_Chairs · 2026-03-02

Reject